# Assessment of Nanopollution from Commercial Products in Water Environments 

**DOI:** 10.3390/nano11102537

**Published:** 2021-09-28

**Authors:** Raisibe Florence Lehutso, Melusi Thwala

**Affiliations:** 1Water Centre, Council for Scientific and Industrial Research, Pretoria 0001, South Africa; flehutso@csir.co.za; 2Department of Chemical Sciences, University of Johannesburg, Johannesburg 2028, South Africa; 3Department of Environmental Health, Nelson Mandela University, Gqeberha 6019, South Africa; 4Centre for Environmental Management, University of the Free State, Bloemfontein 9031, South Africa

**Keywords:** nanopollution, nano-enabled products, engineered nanomaterials, physicochemical properties, aquatic environments

## Abstract

The use of nano-enabled products (NEPs) can release engineered nanomaterials (ENMs) into water resources, and the increasing commercialisation of NEPs raises the environmental exposure potential. The current study investigated the release of ENMs and their characteristics from six commercial products (sunscreens, body creams, sanitiser, and socks) containing nTiO_2_, nAg, and nZnO. ENMs were released in aqueous media from all investigated NEPs and were associated with ions (Ag^+^ and Zn^2+^) and coating agents (Si and Al). NEPs generally released elongated (7–9 × 66–70 nm) and angular (21–80 × 25–79 nm) nTiO_2_, near-spherical (12–49 nm) and angular nAg (21–76 × 29–77 nm), and angular nZnO (32–36 × 32–40 nm). NEPs released varying ENMs’ total concentrations (*ca* 0.4–95%) of total Ti, Ag, Ag^+^, Zn, and Zn^2+^ relative to the initial amount of ENMs added in NEPs, influenced by the nature of the product and recipient water quality. The findings confirmed the use of the examined NEPs as sources of nanopollution in water resources, and the physicochemical properties of the nanopollutants were determined. Exposure assessment data from real-life sources are highly valuable for enriching the robust environmental risk assessment of nanotechnology.

## 1. Introduction

The global commercialisation of nano-enabled products (NEPs) is growing rapidly year on year [1], and it is estimated to grow from USD 39.2 billion in 2016 to over USD 125 billion by 2024 [2]. Approximately 5000 NEPs were identified in various global inventories between 2015 and 2021, belonging to six product categories, namely: health and fitness, electronics and computers, home and garden, appliances, automotive, and food beverages [3,4,5,6,7]. These inventories are generally dominated by health and fitness NEPs, such as sunscreens, personal care products, and clothing products [3,4,5,6,7], which exhibit medium to high probability of emitting engineered nanomaterials (ENMs) into the environment during use, especially water resources (i.e., environmental exposure) [1,6].

Increasing the production and use of NEPs consequently raises the probability of proportional ENMs’ release into aquatic environments; therefore, NEPs are potential sources of daily nanopollution [6,8]. For instance, the release of some commonly applied ENMs in NEPs such as silver (nAg) and zinc oxide (nZnO) into surface water is estimated at approximately 4.9–1700 t/annually [9]. Elsewhere, it was estimated that 50–95% of ENMs (nAg and nTiO_2_) are released into water resources along the life cycle of NEPs [10]. Furthermore, environmental concentrations of ENMs in water systems differ from estimates from in silico studies [11,12,13,14,15]. For example, Ag and Ti’ predicted environmental concentrations (PECs) are reported, respectively, as 0.7–16 µg/L [16,17,18,19], and 0.014–2.2 µg/L [18,19,20], while measured environmental concentrations (MECs) were quantified at 0.03–19.7 µg/L (Ag) [21,22,23], 0.67–150 µg/L (Ti) [12,13,14,24,25,26]. Continuous release of ENMs leads to concentrations of nanopollutants reaching levels that can be hazardous in water resources [27].

In order to address concerns related to nanopollution, a considerable proportion of studies have been undertaken on pristine ENMs [28], but the generated data cannot be directly transferred to ENMs released from NEPs (product-released ENMs) due to differences in physicochemical properties [29]. Differences in physicochemical properties are due to (i) the manipulation of pristine ENMs during preparation for incorporation into NEP, (ii) association of product-released-ENMs with other product components, and (iii) the influence of the NEPs life cycle [28,29,30]. For example, before incorporation into NEPs (e.g., cosmetics), the surface of nTiO_2_ are commonly modified with coating agents such as aluminium hydroxide (Al(OH)_3_) or polydimethylsiloxane (PDMS) to facilitate dispersion in the matrix of NEPs, to prevent/reduce photooxidation and generation of reactive oxygen species (ROS) [31]. The behaviour of such functionalised nTiO_2_ does not resemble pristine ENMs counterparts. Similarly, during the use of NEPs, some of the physicochemical properties of ENMs can be altered after exposure of NEPs (i.e., clothing) to environmental stressors such as ultraviolet and physical forms, the use of fabrics, and how they are washed [32]. For instance, nano-enabled socks that were used released ca 50–100 nm compared to 1–2 nm counterparts released by unused socks [33]. Furthermore, the washing method may also influence the amount and properties of ENMs released from fabrics [34,35].

In that context, scientists focused on examining the environmental risk arising from product-released ENMs, and studies on product-released ENMs and other articles (nanocomposites) have grown from 96 in 2017 [36] to approximately 120 in 2021. The studies illustrate that the concentration of product-released ENMs varies considerably (0.01–35%), and so does the size (<100–385 nm) and other physicochemical properties [36]. Due to the low sample mass/volume attainable after sample preparation and the limited analytical equipment capability to analyse ENMs in complex matrices, fewer studies have optimally characterised product-released ENMs [36,37]. As such, there is a considerable knowledge gap on the exposure characteristics of product-released ENMs, and consequently, robust and realistic risk assessment of product-released ENMs in the environment remains to be established [38,39].

In order to establish and address environmentally realistic risks of product-released ENMs, exposure assessment data need to be strengthened at the various stages of the life cycle of NEPs (production, usage, and end of life) [36,40]. The current study examined the release and exposure characteristics of product-released ENMs from a wide array of NEPs that exhibit a medium to high nanopollution potential toward water resources [6]. The NEPs samples were from the category of health and fitness products: sunscreens, hand sanitiser, body cream, and socks samples. The health and fitness category, specifically personal care products, has been shown to dominate NEPs markets worldwide [3,4,5,6,7]. The selection of NEPs was further influenced by, but not limited to, include a few chemical identities of ENMs, the physicochemical properties of the applied ENMs, and the location of the ENMs within the product; all of which influence the environmental exposure potential of ENMs [6]. By considering the current data gap regarding the environmental risk associated with the use of NEPs, the current study sought to enrich the data on the physicochemical properties of product-released ENMs as an essential component to advance global efforts to determine the probable risk of nanopollutants in aquatic environments.

## 2. Materials and Methods 

Six NEPs, namely three sunscreens, SUN1 (nTiO_2_ + nZnO), SUN2–3 (nTiO_2_); body cream CA1 (nTiO_2_ + nAg); sanitiser; SAN1 (nAg); and socks SK1 (nTiO_2_ + nAg) were purchased from South African retailers. The physicochemical properties of the ENMs incorporated in the six NEPs varied and were previously reported [41]. Briefly, SUN1 contained elongated nTiO_2,_ and angular nZnO particles sized 14 × 62 nm and 35 × 38 nm, respectively. SUN2 and SUN3 contained angular-shaped nTiO_2_ with a size range of 20–28 × 27–32 nm and 20–28 × 27–32 nm, respectively. Near-spherical nAg particles with the size range of 22–37 nm were observed in SAN1. CA1 and SK1, respectively, contained nTiO_2_ particles sized at 8 × 53 nm (elongated) and 32–203 × 48–135 nm (angular), the NEPs also contained near-spherical nAg ranging 18–28 nm. The particles were negatively charged, and the nTiO_2_ phase was rutile or anatase. The total amounts of ENMs differed between NEPs [41].

### 2.1. Procedures for ENMs Release 

The procedures used to investigate the release of ENMs from NEPs differed and are described in Section 2.1.1 and Section 2.1.2. In all instances, the procedures were adapted to simulate conditions (but not fully replicate actual life cycle stages) that promote ENMs release from the NEPs. However, some stages applicable during the use of current NEPs (e.g., application of sunscreen or sanitiser to the skin, prior wearing of socks) were eliminated as the primary focus of the study was the analytical determination of the potential for ENMs release. Furthermore, simple (or standard) aqueous media were preferred to avoid physicochemical complexation, which occurs when using complex media and uncharacterised commercial detergents [42]; however, release evaluations in complex media with greater environmental realism are the cornerstone of nanotechnology risk assessment and are recommended for future studies as analytical capabilities and access advances. All investigations were carried out in triplicate. 

#### 2.1.1. Release of ENMs from Suncreen 1–3 (SUN1–3) and Body Cream 1 (CA1)

The release of ENMs from SUN1–3 and CA1 followed a slightly modified protocol of Botta et al. [43]. Briefly, 2 g of SUN1–3 and CA1 were aged in 180 mL of release media for 48 h at 25 °C under darkness (plastic beakers were capped and covered with heavy-duty aluminum foil) and under illumination: −6000 lux (uncapped transparent plastic beakers). The SUN1–3 ENMs were released in Milli-Q water (18 MΩ·cm), freshwater, seawater or swimming pool water (S1.1), while the CA1 ENMs were released using Milli-Q water only; the properties of the release media are given in S1.1. ENMs were released by agitating the suspension at 400 rpm for 48 h, and the sample volume was maintained by continuously adding aqueous media throughout the 48 h. Agitation was stopped after 48 h, and the samples were allowed to settle for another 48 h, a step that caused sedimentation that resulted in two phases (surface suspension and sediment); the overall duration of ENMs release was 96 h. The two phases were separated by sampling 150 mL of stable suspension; the sediments were not disturbed during sampling. The stable suspensions were prepared for product-released ENMs analysis (Section 2.2).

#### 2.1.2. Release of ENMs from Sock1 (SK1) and Sanitiser1 (SAN1)

The release of ENMs from SK1 was undertaken by adapting previously developed methods [44,45]. Briefly, areas (spots/regions) of the sock material marketed and experimentally confirmed to be incorporated with nAg and nTiO_2_ [41] were cut from SK1 samples and transferred into 1 L glass bottles, washed with 200 mL of the release media (Milli-Q water, tap water, and sodium dodecyl sulfate as a detergent). The detergent media was prepared in two ways: (i) sodium dodecyl sulfate 1, which was prepared in Milli-Q water, while (ii) sodium dodecyl sulfate 2 was prepared in tap water. The samples were washed by shaking at 350 rpm at 40 °C for 12 h (2 washes). After the final washing cycle, the fabrics were removed from the washing water, and the samples were prepared for analysis (Section 2.2).

For SAN1, ENMs were released following a slightly modified method of Benn et al. [46] and Mackevica et al. [47]. Briefly, 1 mg/L of the sample was prepared in Milli-Q water and ENMs released by agitating at 350 rpm at 40 °C for 24 h. After ENMs’ release, samples were prepared for analysis (given in Section 2.2). 

SUN1 (nZnO), CA1 (nAg), SAN1 (nAg), and SK1 (nAg) were incorporated with ENMs that are relatively soluble. The released ions were recovered from the release media through sequential filtration. The samples were sequentially filtered using Amicon^®^ Ultra-15 30 K centrifugal filters (30000 MWCO, Merck, South Africa), followed by further centrifugation using Amicon^®^ Ultra-15 3 K centrifugal filter devices (3000 MWCO, Merck, South Africa) for 30 min at 10,000 rpm for each filtration step. The released ions (filtrates from the 3 K centrifugal filter device) were quantified (Section 2.2.3).

### 2.2. Physicochemical Properties of Product-Released ENMs

#### 2.2.1. Electron Microscopy

Images of product-released ENMs were obtained using a JEOL-JEM 2100 high-resolution transmission electron microscope coupled to energy-dispersive X-ray spectroscopy (HR-TEM-EDX) (Tokyo, Japan) fitted with a LaB6 filament operated at 200 kV. A Cu grid with a holey carbon film was dipped in the sample solution and air-dried for 12 h, followed by TEM-EDX analysis. Multiple images were captured at different spots on the grid to measure the product-released ENMs’ size (minimum particle set at 50) using the ImageJ software. 

#### 2.2.2. Surface Charge of Product-Released ENMs

A Zetasizer Nano ZS (Malvern Instruments, Worcestershire, United Kingdom) was used to determine the zeta (ζ) potential of product-released ENMs in the release media, which measured the physicochemical properties reported in Appendix A.

#### 2.2.3. Elemental Analysis

Elemental analysis of product-released ENMs was performed using inductively coupled plasma mass spectrometry (ICP-MS, Icap Q, Thermo Fisher Scientific, Waltham, United States of America). For total Ti, Zn, Ag, and Si analysis, samples were predigested following a modified MARS 6 Method Note Compendium [48]. Product-released ENMs samples were transferred into digestion vessels, 5.0 mL of nitric acid (HNO_3_) (70%, Merck, Johannesburg, South Africa) was added, followed by swirling the vessel and leaving it open for approximately 10 min. After 10 min, 2.0 mL of hydrogen peroxide (H_2_O_2_) (37%, Merck, Johannesburg, South Africa) and hydrofluoric acid (HF) (49%, Merck, Johannesburg, South Africa) were added to samples containing Ag and Ti, respectively. The microwave digestion program followed the cosmetic and textile heating program highlighted in the MARS 6 Method Note Compendium [48]. All product-released ENMs’ digests were filtered using a 0.45 µm filter syringe (Merck, Johannesburg, South Africa) and prepared for ICP-MS analysis, monitoring ^66^Zn, ^48^Ti, ^107^Ag, ^28^Si, ^27^Al, and ^45^Sc (internal). The performance of the digestion method was evaluated by digesting both bulk (Zn, Ti and Ag, Anatech instruments, Johannesburg, South Africa) and nTiO_2_ (Tavo commercial nanocomposite, Merck, Johannesburg, South Africa), Ag (bare and aminated, nanoComposix, San Diego, United States of America) and ZnO (Z-cote, a commercial nanocomposite, BASF, Johannesburg, South Africa). The recovered filtrates were acidified (5% using HNO_3_) and directly analysed for the dissolved ions. Appropriate sample dilutions were performed prior to analysis. 

### 2.3. Data Analysis

Statistical analysis and drawing of graphs were performed using GraphPad Prism8 version 8.4.3 for Windows (GraphPad Software, La Jolla, San Diego, United States of America). Student’s t-test and two-way ANOVA with Tukey’s HSD post hoc test were applied to examine differences between samples, with significance tested at α = 0.05.

## 3. Results and Discussion

### 3.1. Characterisation of Product-Released ENMs

The release methods used in the current study successfully released ENMs from all NEPs as confirmed in all exposure media variants (Section 3.1.1, Section 3.1.2 and Section 3.1.3).

#### 3.1.1. Sunscreen 1–3 (SUN1–3) Product-Released ENMs 

Product-released ENMs in different release media (Milli-Q, freshwater, swimming pool water, and seawater) obtained under light conditions are provided in Figure 1 (TEM images) and Appendix A (elemental profile). TEM images and elemental profiles of product-released ENMs obtained under dark conditions are given in Appendix A, respectively. Variation of illumination conditions and release media did not influence the morphology of product-released ENMs. SUN1-released nTiO_2_ were elongated, while nZnO was angular in shape. SUN2–3-released nTiO_2_ were angular in shape, shapes that were previously reported [41]. 

Product-released ENMs were still predominantly associated with aluminium (Al) and silicon (Si) (Appendix A), indicative of remnants of coating agents [49,50,51], either intact on the surface of product-released ENMs or in the release media. The intensities of the Al and Si peak varied between the release media and the type of sunscreen (Appendix A), demonstrating that the release media or exposure conditions affected the ENMs coating agents differently.

The findings confirmed that product-released ENMs were not released in naked forms (pristine ENMs), supporting previous reports that product-released ENMs are commonly released associated the matrix of NEPs (transformed state) [14,42]. For example, in SUN1, Si appeared to have been predominantly released into the media, while Al partially remained adsorbed in product-released nTiO_2_ (Appendix A). In SUN2 and SUN3, Si remained mainly attached to product-released nTiO_2_ (Appendix A). The desorption of the coating agents from product-released ENMs surface has implications for the exposure potential of ENMs in aquatic environments, as they influence their reactivity [52], bioavailability and toxicity to aquatic organisms [53,54,55]. The findings partly illustrated that the environmental exposure characteristics arising from the use of NEPs could not be accurately established from studies using pristine ENMs, which leads to the need for refinement and standardisation of ENMs release protocols to improve exposure assessment data. 

Similar to the shape of the released ENMs, the sizes (width × length) were also unaffected by both illumination and release media (Table 1 and Appendix A). The average size of SUN1-released nZnO was 32–36 × 32–40 nm, while product-released nTiO_2_ were 7–9 × 66–70 nm. The SUN2-released ENMs sizes were 27–30 × 33–37 nm, while SUN3-released ENMs were 21–22 × 25–28 nm. The average size of SUN2-released ENMs was in agreement with previously reported sizes [41]. The distribution of the SUN1-released nZnO particles (W × L) in Milli-Q water and freshwater were comparable, and the distribution densities were similar to the ENMs incorporated in SUN1. SUN1-released nZnO distributions (W × L) in seawater and swimming pool water were similar; the distribution (upper and lower quartiles and violin density) was comparable to the ENMs found in SUN1. The particle distributions of SUN1-released nTiO_2_ (W × L) were similar in all media. The upper and lower quartiles of the SUN1-released nTiO_2_ (W × L) distribution slightly varied; the ENMs in SUN1 but were comparable in violin density [41]. In all release media, the product-released nTiO_2_ distribution profiles of SUN2 and SUN3 were also similar and generally comparable to ENMs in the respective SUNs. While the distribution profiles in all SUN’s were comparable to ENMs in NEPs, few exceptions, especially on violin shape/structure and quartiles, were observed.

Product-released ENMs of all sunscreens were negatively charged, illustrating that, as expected, illumination did not influence the surface charge (under light: Figure 2 and dark: Appendix A). Although all product-released ENMs were negatively charged, the stability of product-released ENMs varied between the release media. Relatively high ζ potentials (negative or positive, a minimum value of 22 mV) are considered electrically stable, while lower ζ potentials are less stable and can lead to rapid agglomeration of nanoparticles [56]. All sunscreen-released ENMs were stable in Milli-Q and freshwater and unstable in seawater and swimming pool water. The difference in the stability of sunscreen-released ENMs is well corroborated with the TEM-EDX results (Appendix A), where the coating agents of product-released ENMs were affected differently by the different release media. ENMs are functionalised with coating agents to improve stability [57]; therefore, alteration of the ENMs coating agents will directly affect the stability of ENMs and their fate in aquatic systems [58,59]. 

The findings of the current study were comparable to previous reports. For example, the size range of elongated product-released nTiO_2_ obtained in the current study was comparable to the range (10 × 139 nm) of sunscreen-released nTiO_2_ (elongated) previously reported [42,43]. The negative surface potential of sunscreen-released ENMs was also previously reported [43,60].

#### 3.1.2. Sanitiser 1 (SAN1) and Body Cream (CA1) Product-Released ENMs 

SAN1 and CA1 ENMs were successfully released into the respective media, as shown in Figure 3 and Appendix A (size distribution of product-released ENMs). SAN1-released nAg were near-spherical and averaged 10 ± 2 and 23 ± 4 nm, indicating distinct size classes. The SAN1-released nAg generated two distribution profiles that differed in the upper quartiles; one of the profiles was comparable to the ENMs in SAN1 [41]. The other distribution differs from the ENMs profile in SAN1 on width, indicating possible agglomeration. The SAN1-released nAg ζ potentials were determined to be −32.5 ± 2.1 mV.

Binary CA1-released nTiO_2_ and CA1-released nAg were detected under both illumination conditions (Figure 4 and Appendix A), the Si peak of the coating agents was also detected. The CA1-released nTiO_2_ were elongated in shape and had an average size of 8 ± 3 × 60 ± 13 nm (under light) and 9 ± 3 × 66 ± 9 nm (under dark), indicating that the size was not affected by variation in illumination. Near-spherical CA1-released nAg were detected in three distinct average sizes of 12 ± 3, 27 ± 7, and 49 ± 9 nm under light conditions, relative to 10 ± 3, 28 ± 8, and 54 ± 8 nm under dark conditions, indicating that illumination variation did not affect ENMs sizes. The distribution and the violin density of CA1-released nAg obtained under light and dark conditions were similar. The distribution density of the CA1-released nAg and ENMs was comparable, but differed in the upper quartiles, indicating possible particle transformation. Similarly, the distribution of CA1 released nTiO_2_ was comparable, except in the lower quartiles of CA1 released nTiO_2_ obtained under dark conditions. CA1-released ENMs obtained under light and dark conditions were negatively charged at −23.6 ± 1.3 and −22.8 ± 1.2 mV, respectively. 

The presence of different product-released nAg size classes indicated that the ENMs were transformed during release, since the primary size of the ENMs incorporated in the NEPs averaged 21.7 ± 6 (CA1) and 22 ± 7 nm and 37 ± 4 nm (SAN1) [41]. In aquatic environments, pristine nAg are susceptible to undergo various transformations [61], including oxidative dissolution and reformation of Ag particles, leading to the formation of particles of different sizes [62,63]. Peretyazhko et al. [64] found that after the dissolution of pristine nAg, the size of the particles increased due to Ostwald ripening. In the case of particle size decrease, some studies attributed the reduction to the dissolution of nAg, followed by the reduction-driven formation of smaller nAg [65,66]. Furthermore, it was shown that in the absence of environmental factors such as ultraviolet radiation and environmental ligands, a simple dilution of concentrated nAg suspensions and colloidal Ag-based products such as SAN1 can cause particle destabilisation leading to the formation of agglomerates and the reduction in particle size [67,68]. The change in the particle size of nAg incorporated into body cream and mouth spray in artificial sweat and saliva was previously reported [69]; product-released ENMs experienced significant growth in size from 5 to 25 nm to 10 to 800 nm.

Environmental exposure to product-released ENMs in aquatic environments has been reported mainly from commercial clothing [70,71], personal care products (toothbrushes, toothpaste, face masks, shampoo, and detergents) [46,47], and paints [72]. The sizes of personal care-released nAg and paint-released nAg were 42–500 nm [46,47] and <15–100 nm [72], respectively. Similar to most release studies, the product-released nAg were still embedded in the NEPs’ matrix. Herein, the SAN1-released nAg did not appear to be embedded in the product matrix and were individually isolated or agglomerated; such findings further illustrate that ENMs release potential is influenced by their *loci* in products and product formulation. CA1-released ENMs were often visualised to be encircled by a layer that could not be accurately identified, whether being components of the NEP’s matrix or ENMs coating agents; however, it is worth noting that Si was detected in the sample by EDX (Figure 4). The physicochemical state at which the product-released ENMs were detected in aqueous environments was predominantly related to the matrix of NEPs. For example, SAN1 was a clear liquid suspension with a viscosity comparable to water, while CA1 was a semi-solid cream made up of organic compounds.

#### 3.1.3. Socks 1 (SK1) Product-Released ENMs

Washing SK1 released binary ENMs (nTiO_2_ and nAg) (Figure 4). SK1-released nAg were near-spherical and angular in shape and averaged 8 ± 4 nm and 21–76 × 29–77 nm, respectively. The angular particles were smaller and rapidly agglomerated. SK1-released nTiO_2_ were angular and averaged 80 ± 25 × 79 ± 29 nm (Appendix A). The distribution of SK1-released nTiO_2_ and the spherically shaped SK1-released nAg and the respective ENMs in SK1 are comparable. The profile of angular/irregular shaped SK1-released nAg and nAg in SK1 slightly differs, an expected observation since nAg size was affected by the ashing procedure [41]. 

SK1-released ENMs were coated with Si, and Al, and the coating agents were found to be intact on some SK1-released ENMs (Appendix A). Similar to the previous product-released ENMs (in the preceding sections), the SK1-released ENMs’ surface was negatively charged (−33.0 ± 2.1 mV). The current findings are in agreement with previous reports, whereby product-released nAg (20–40 nm) and product-released nTiO_2_ (60–350 nm) were detected after washing nano-enhanced textiles [70,71].

It is worth mentioning that considerable analytical challenges were initially experienced during the characterisation of SK1-released ENMs. First, SK1-released ENMs were not detected (TEM-EDX) without a pre-enrichment step, especially for SK1-released nAg. After sample enrichment, small particles (~4–6 nm) were imaged but could not be identified because the EDX beam rapidly destroyed them. Finally, the washing detergent introduced a thick layer that concealed the SK1-released ENMs underneath (Appendix A). To improve TEM-EDX characterisation, the number of SK1 units washed concurrently was increased; for this part, the release media was limited to Milli-Q water. Increasing the number of SK1 samples washed simultaneously and concentrating the sample through centrifugation improved TEM-EDX characterisation and enabled SK1-released ENMs particle size quantification. 

Overall, the characterisation of product-released ENMs showed that all NEPs investigated in the present study are potential nanopollution sources for water resources. The shapes of the respective product-released ENMs were similar to the ENMs incorporated into the respective NEPs, whose physicochemical properties were previously reported [41]. The sizes of SUNs-released ENMs were comparable to the sizes determined in the NEPs [41]. However, in the case of CA1, SAN1, SK1, the product-released ENMs sizes were slightly different from the ENMs incorporated into the NEPs [41], especially for nAg, where the transformation occurred in terms of the change in particle size (increase and decrease). The physical properties of product-released ENMs are crucial in understanding the behaviour, fate and effects of nanopollutants in aquatic environments, where several studies have already reported their presence in real environmental samples [13,14,73,74,75].

### 3.2. Elemental Quantification of Product-Released ENMs

The digestion, analysis, and recovery method of nano- and bulk reference standards were within the acceptable ranges of (75–107%) Ti, (72–97%) Ag, (74–98%) Zn, (70–91%) Al and (70–87%) Si. 

#### 3.2.1. Sunscreen 1–3 (SUN1–3) Product-Released ENMs 

The total concentration of Ti, Zn, and Zn^2+^ released relative to the initial amount of ENMs added to the sunscreens varied and ranged in general between 0.4 and 8% (*w*/*w*) (Figure 5). SUN1–3 released Ti at different extents; in most exposure scenarios, SUN3 > SUN2 > SUN1. In addition to Ti release, SUN1 simultaneously released Zn and Zn^2+^ in the range of 0.67–5.7% (*w*/*w*) and 0.5–3.0% (*w*/*w*), respectively (Figure 6). The amounts of Zn and Zn^2+^ in the respective product-released ENMs release media were mostly different (Figure 6). Indicative that SUN1 generally releases Zn in particulate and ionic forms.

The amounts of Ti, Zn, and Zn^2+^ released from sunscreens were influenced by nature of the NEPs formulation (the initial amount present in the NEP matrix and the product matrix) and simulated environmental conditions (water chemistry and variation in illumination). The influence of the initial amount present in the NEPs was observed between SUN2 and SUN3 (being of the same brand). SUN3, which contained more nTiO_2_ [1.6% (*w*/*w*)] compared to SUN2 [0.95% (*w*/*w*)] [41], released relatively higher amounts of Ti (*p* = 0.0001–0.01). A further comparison of the amount of Ti released by SUN1–3 showed that the NEPs matrix also influenced Ti release. Although SUN1 contained relatively more nTiO_2_ [4.31% (*w*/*w*)] than SUN2–3 [<3% (*w*/*w*)] [41], the total amounts of Ti released from SUN1 were lower than SUN2–3 (Figure 6)—probably an influence of the formulation of the product on the release of ENMs. 

In terms of environmental conditions, the amounts of Ti, Zn, and Zn^2+^ were mainly influenced by water chemistry rather than illumination variations; illumination rarely influenced the amounts released. In descending order, the amount of Ti released from SUN1 under light and dark conditions was Milli-Q water ≥ freshwater ≥ seawater > swimming pool water and freshwater > Milli-Q water > Seawater ≥ swimming pool water, respectively. In the case of Zn, the trend of the amounts released under light and dark conditions, in descending order, was Milli-Q water > freshwater> seawater > swimming pool water and Milli-Q water > seawater > freshwater > swimming pool water, respectively. The amounts of Zn^2+^ followed a descending order of Milli-Q water > seawater > freshwater ≥ swimming pool water for both illuminations. The Ti amount trends (descending order) of SUN2 and SUN3 were Milli-Q water > freshwater ≥ swimming pool water > seawater and Milli-Q water > swimming pool water > freshwater ≥ seawater for both illuminations, respectively. 

The ionic strength of the release media probably enhanced the agglomeration and sedimentation rate, thus probably causing the differences in the amount released. The release media influenced the dispersion of the sunscreens in the media was different; for example, in Milli-Q water and freshwater, the sunscreens dispersed thoroughly and turned into a homogeneous milky solution, while in other cases, the sunscreen matrix fragmented and formed flocculates. The difference in the dispersion and sedimentation of the NEPs matrix in the release media has implications for ENMs’ exposure dynamics in the aqueous phase, as the two (uniform mixture and flocculates) will have different sedimentation rates; flocs sediment faster due to gravity [76]. Different ENMs sedimentation rates were reported in different water chemistries and are influenced by ionic strength, ionic species, and dissolved oxygen [77].

Overall, the current findings illustrated the varying nanopollution characteristics arising from sunscreen NEPs in different water quality environments and that the degree of nanopollution depends on both the NEPs’ matrix properties and recipient resource water quality. Furthermore, the results showed that the product-released ENMs will pollute not only the aqueous phase of aquatic environments but also sediments, in addition to adsorption to abiotic and biotic entities. The sedimentation rate influenced the concentrations detected in the suspension, a factor that will be at play in real water bodies as driven by the velocity of the water and other characteristics. Investigations of ENMs sedimentation were carried out on pristine ENMs, and it was found [78] that 50% and 70% of nTiO_2_ and nZnO were found to sediment within the first 24 h and continued to slowly sediment for the next 2 to 14 days in natural water, respectively. Similarly, the study by Botta et al. [43] showed that a significant proportion of sunscreen-released nTiO_2_ in seawater aggregated and sedimented. The rate of sedimentation influences the exposure dynamics of benthic organisms. Beyond the release stage, the behaviour of product-released ENMs in aquatic environments and the effects on benthic organisms are not well understood and warrant detailed attention. As such, at more robust levels, ENMs exposure assessment must consider aquatic resource characteristics.

#### 3.2.2. Sanitiser 1 (SAN1) and Body Cream (CA1) Product-Released ENMs

The amount of Ag, Ag^+^, and Ti released from SAN1 and CA1 varied (Figure 6). SAN1 released considerably higher amounts of Ag than CA1 (*p* = 0.001); the characteristics of the NEPs matrix probably caused the observed difference—further illustrating the influential role of the NEPs matrix in the potential for exposure to ENMs. Both SAN1 and CA1 released Ag in particulate and dissolved forms. The amount of Ag and Ag^+^ in the respective release media of SAN1 (*p* = 0.0002) and CA1 (*p* = 0.003–0.005) varied, indicating the coexistence of particulate and ionic Ag. CA1 further released Ti amounts higher and comparable to Ag under light (*p* = 0.02) and dark (*p* = 0.056) conditions.

Some NEPs containing nAg were classified as having medium to high exposure potential to water resources [6,7,79]. The studies reported that toothbrushes released nAg (5.9–626 ng/L) [47], paints released nAg (30%) [72], and plush toy exterior fur released nAg (<1–35%) [80]. It is estimated that Ag can be released from products in the range of 25–100% in wastewater treatment plants [81]. In most cases, the NEPs release Ag in the particulate or ionic form at varying degrees [47,80]. The form and extent of Ag release from nAg are complex because speciation is influenced by various factors, such as particle size, coating agents, and release media characteristics [82,83,84,85,86]. 

The dissolution of nAg from both NEPs may be due to the small-sized particles incorporated in SAN1 (10–37 nm) and CA1 (13–44 nm) and the change in particle size of product-released nAg (as observed by the detection of particles of different sizes) may have contributed to the degree of dissolution observed in the exposures of SAN1 and CA1. Nanoparticle size reduction was previously reported to result in increased dissolution due to increased surface area [87,88].

#### 3.2.3. Socks 1 (SK1)-Released ENMs

SK1 released 0.004–0.100 mg/L and 2.66–5.98 mg/L of total Ag and Ti, respectively (Figure 7). The Ag and Ti concentrations released from SK1 were not normalised back to the initial concentration incorporated into the NEPs because the Ag and Ti present in different SK1 materials were inconsistent. Incorporation of ENMs of different properties by manufacturers has recently been reported [89]. As illustrated in Figure 7, the amounts quantified in the different wash cycles varied between the release media and fractions. The amounts quantified for particulate fractions in Milli-Q water were not different (*p* = 0.22–0.67). In the case of tap water, the difference was only observed in the >0.45 µm fraction, where a higher Ag concentration was determined in the second cycle. The amount released from sodium dodecyl sulfate 1 and sodium dodecyl sulfate 2 also varied; relatively large amounts were detected in the first cycle for >0.45 µm (*p* = 0.001–0.03) and <0.45 µm (0.06–0.21). For Ag^+^, only sodium dodecyl sulfate 2 released higher amounts in the first cycle (*p* = 0.01). Released Ag^+^ was detected in comparable amounts between cycles in Milli-Q water, tap water, and sodium dodecyl sulfate 1 (*p* = 0.27–0.99). As shown in Figure 7, sodium dodecyl sulfate 1 release media mainly affected Ag forms, compared to tap water and especially Milli-Q water. 

In cases where Ag amounts varied in different fractions, more Ag was detected in >0.45 µm, while <3 kDa was comparable or lower than <0.45 µm. Contrary to Ag and Ag^+^, the amounts of Ti were comparable in the first and second wash cycles for all wash media, except for sodium dodecyl sulfate 1, where the amounts of Ti were higher in the first cycle. As shown in Figure 7, the amounts of Ti in the different fractions were comparable, except for tap water, sodium dodecyl sulfate 1 and sodium dodecyl sulfate 2, where the highest amounts were quantified in fractions> 0.45 m and fractions >0.45 µm and <0.45 µm fractions, respectively. Overall, nAg release was more affected by the simulated washing conditions than nTiO_2_ incorporated in SK1.

The environmental exposure of particulates and ions of Ag [33,34,35,44,90,91] and Ti [71,92], respectively, released from commercial textile products enabled with nAg or nTiO_2_ have been investigated, although, in some instances, there were differences between studies. The differences were mainly caused by the assessment of different clothing materials, the type of ENMs nanocomposite, the initial amounts of ENMs added to the NEPs, ENMs incorporation methods, ENMs location within the NEPs, and the chemistry of the release media. 

Nonetheless, the current study correlated with some previous reports on the high amount of Ag detected mainly in the first wash cycle [14,75,77,80] and the detection of higher amounts of particulate Ag [44]. Thus, elevated ENMs release from fabric NEPs can be expected from initial washes after purchase. Overall, the amounts released in the current study were in agreement with the previous reporting of 0.32–38.5 mg/L for Ag [34,91,93] and 5 mg/L for Ti [71]. Exposure assessments of commercial TiO_2_ nano-enabled textiles compared to nAg-enabled textiles are currently scarce. The high market penetration of nAg functional textiles and the primary function of nAg (antimicrobial properties) could be the reason behind the difference in the number of studies undertaken. 

#### 3.2.4. Release of ENMs Coating

ENMs coating agents in SUN1–3 (Appendix A), CA1 (Appendix A) and SK1 (Appendix A) were somewhat desorbed from the surface of ENMs, and the components of the coating agent (Si and Al) were released into the respective media. The Si and Al coating agents of the released ENMs were determined in <3 kDa filtrate to avoid coating agents still attached to the product-released ENMs. Although TEM-EDX analysis showed that Si and Al were coated on the surface of ENMs, the presence of these elements as part of the other matrix of NEPs may exist; for all NEPs, manufacturers neither declared the element nor the quantity. Because of uncertainties, for this exercise, the Si and Al are assumed to originate from ENMs coatings and the overall NEPs matrix. 

The findings on the extent to which ENMs coating agents were released were recorded in Milli-Q water release media. The amount of Si released varied between the NEPs (Table 2). 

Si was detected in the respective release media (SUN1–3 and CA1) in the descending order of CA1 > SUN1 > SUN3 ≥ SUN2 under both illumination conditions. The SUN1 ENMs, which were coated with Si and Al, released Si in large amounts compared to Al; Al amounts in the product-released ENMs media of SUN1 were below the detection limit (LOD = 10 µg/L). Elemental concentrations were consistent with the EDX observations, where Si desorbed and released into the product-released ENMs release media, while Al was still partially attached to the product-released ENMs (Appendix A). Similarly, the low amounts of Si detected in SUN2 and SUN3 corroborated the earlier findings of TEM-EDX (Appendix A), where Si was observed to be still attached to product-released ENMs. In the case of CA1, which released the highest amounts of Si, TEM-EDX analysis (Appendix A) showed that most of the Si disassociated from product-released ENMs. For SK1, the amounts of Si and Al were determined to be 10 and 4.76 mg/L, respectively, also confirming the release observed with TEM-EDX (Appendix A). 

Although ENMs incorporated into NEPs are well known to be enclosed with coating agents [94] and have been shown to be altered during the aqueous ageing of nanocomposites (used in cosmetics) and released into aquatic environments [52,95,96,97]; the amount of the coating agent components released from NEPs is often not reported. Until recently, nanocomposites intended for NEPs formulations, such as sunscreens, were evaluated [97]; 1.5–2% (*w*/*w*) of Si was released into ultrapure water, while higher amounts of 88–98% (*w*/*w*) were simulated freshwater and seawater. It is imperative that the amounts of released coating agents are quantified and considered when the risks of product-released ENMs are evaluated, as it is currently unclear whether the components of the coating agent influence the product-released ENMs to what extent, and therefore future studies should evaluate their association.

## 4. Conclusions

The study successfully illustrated the nanopollution of water media during the simulated use of NEPs by characterising product-released ENMs from a wide range of products. The product-released nTiO_2_ were elongated (7–9 × 66–70 nm) or angular (21–80 × 25–79 nm) in shape; product-released nAg were near-spherical (12–49 nm) or angular (21–76 × 29–77 nm) and product-released nZnO were angular (32–36 × 32–40 nm) in shape. The ENMs release rate was determined to be *ca* 0.4–95% relative to the initial amount of ENMs added to NEPs. The extent and characteristics of product-released ENMs were influenced by receiving water quality, ENMs *loci* in the product, and the formulation of the product matrix, while illumination variation essentially did not exert influence. Predominantly, the product-released ENMs were released in association with coating agents (Si and Al) and ionic forms. Considering the influential role the surface coating exerts on the behaviour and toxicity of ENMs in water resources, we highly recommend the reporting of the presence and characteristics of coating agents on product-released ENMs since it is currently not standard practice.

SUN1, CA1 and SK1 released binary ENMs. Typically, there is currently limited information on the environmental implications of ENMs mixtures, more so for product-released ENMs; hence, we encourage more studies to unravel the exposure and hazard dynamics of product-released ENMs mixtures.

Nanopollution is an emerging environmental health issue that is yet to be clearly quantified. Nevertheless, proactive mitigation measures can reduce environmental exposure, for instance, the reduction in ENMs quantity in NEPs (safety-by-design principle), since this study demonstrated that the NEPs sample caused nanopollution. In low- and middle-income countries, such as South Africa, where the current study was carried out, there must be accelerated efforts to estimate the size of the NEPs market to refine the extent of nanopollution, as developed regions have advanced in that aspect.

## Figures and Tables

**Figure 1 nanomaterials-11-02537-f001:**
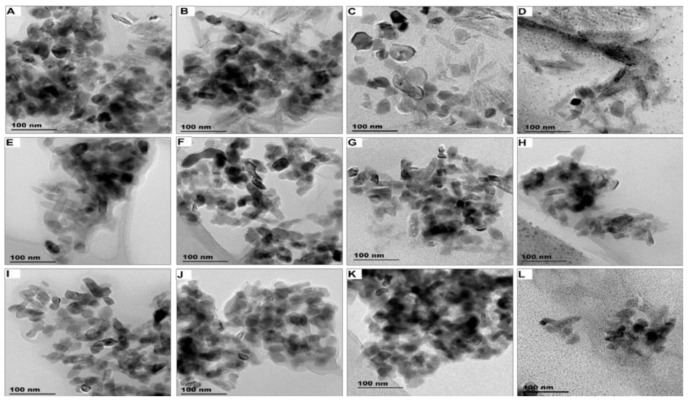
TEM images of product-released ENMs obtained under light conditions for SUN1 detected in Milli-Q water (**A**), freshwater (**B**), swimming pool water (**C**), seawater (**D**), SUN2 detected in Milli-Q water (**E**), freshwater (**F**), swimming pool water (**G**), seawater (**H**) and SUN3 detected in Milli-Q water (**I**), freshwater (**J**), swimming pool water (**K**), seawater (**L**).

**Figure 2 nanomaterials-11-02537-f002:**
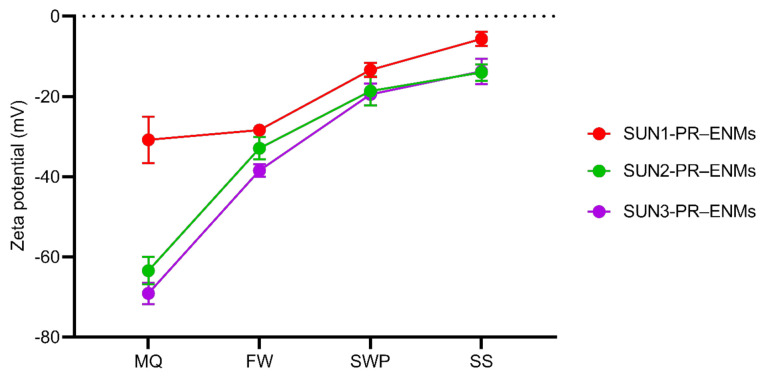
Zeta potential of product-released ENMs (PR–ENMs) obtained under light conditions in different release media of Milli-Q water (MQ), freshwater (FW), swimming pool water (SWP), and seawater (SS).

**Figure 3 nanomaterials-11-02537-f003:**
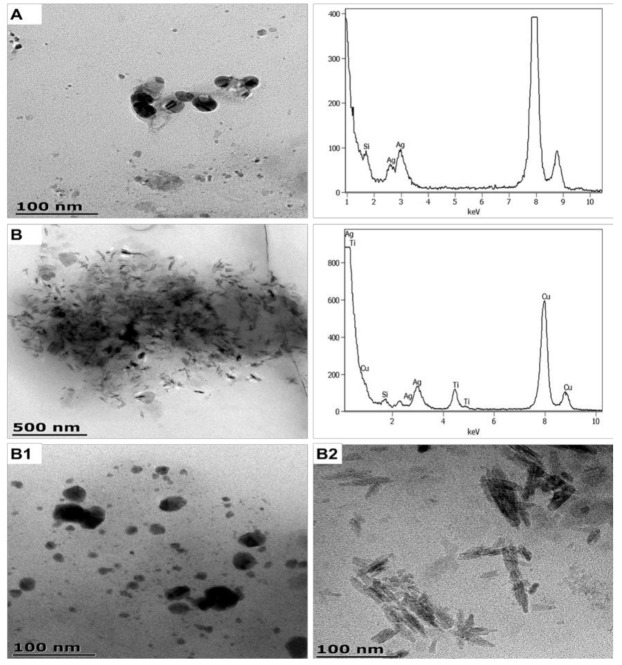
TEM-EDX illustrating SAN1-released ENMs (**A**) and binary CA1-released ENMs obtained under light conditions (**B**). (**B1**,**B2**) are higher magnification of image B showing product-released nAg and product-released nTiO_2_, respectively.

**Figure 4 nanomaterials-11-02537-f004:**
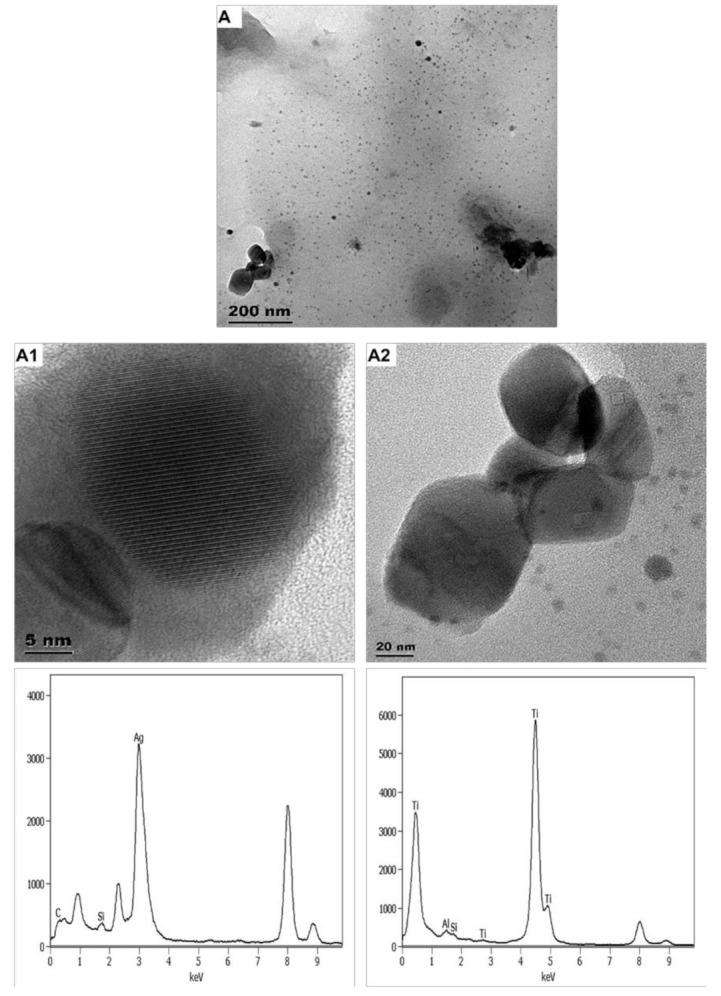
Images and respective spectra obtained from TEM-EDX characterisation of SK1-released ENMs (**A**). Images (**A1**,**A2**) are high magnification of image A, specifically showing near-spherical SK1-released nAg and angular SK1-released nTiO_2_ particles, respectively.

**Figure 5 nanomaterials-11-02537-f005:**
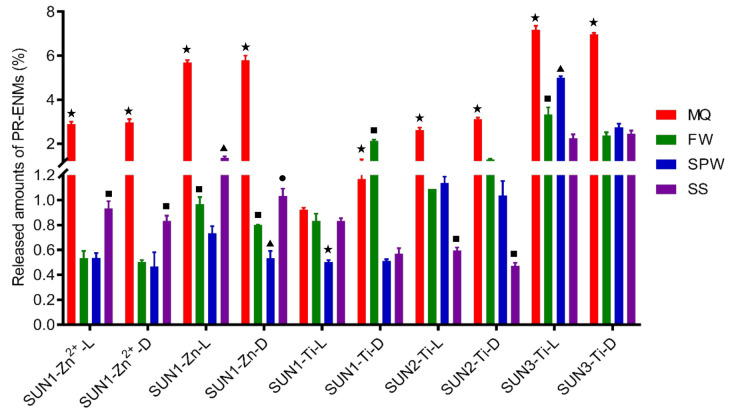
The amount of Zn^2+^, Zn, and Ti released from SUN1–3 in different release media (Milli-Q water (MQ), freshwater (FW), swimming pool water (SPW), and seawater (SS) under light (L) and dark (D) conditions.The differing of symbols (★ ■ ▲) on top of error bars indicates statistical difference (*p* < 0.05) between the release media treatments.

**Figure 6 nanomaterials-11-02537-f006:**
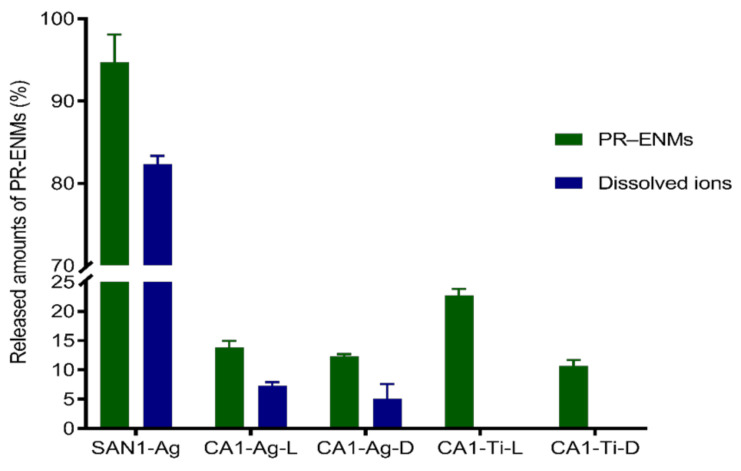
The amounts of SAN1 and CA1-released nAg and released Ag ions; L and D denote light and dark conditions, respectively.

**Figure 7 nanomaterials-11-02537-f007:**
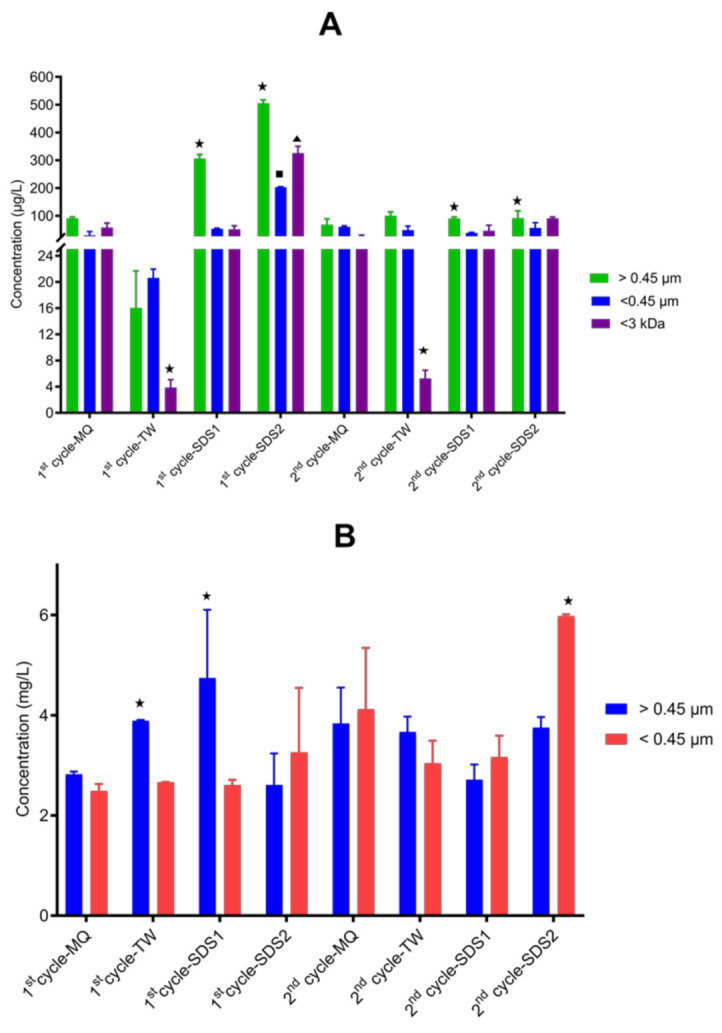
Total amounts of Ag (**A**) and Ti (**B**) released from SK1 in Milli-Q water (MQ), tap water (TW), sodium dodecyl sulfate 1 (SDS1), and sodium dodecyl sulfate (SDS2) in two wash cycles. The symbol (★ ■) on top of error bars denote statistical difference (*p* < 0.05) between the released Ag and Ti fractions per wash cycle.

**Table 1 nanomaterials-11-02537-t001:** Particle shape and the average size of product-released ENMs in different release water media. ^a^ and ^b^ are the sizes of product-released ENMs obtained under light and dark conditions, respectively.

Release Media			Milli-Q Water	Freshwater	Swimming Pool Water	Seawater
Sample	ENMs Type	ENMs Shape	Size (nm)	Size (nm)	Size (nm)	Size (nm)
SUN1	ZnO ^a^	Angular	34 ± 6 × 37 ± 7	35 ± 9 × 32 ± 8	38 ± 7 × 39 ± 5	36 ± 4 × 40 ± 7
	ZnO ^b^	Angular	37 ± 9 × 39 ± 8	36 ± 4 × 37 ± 4	37 ± 9 × 39 ± 9	43 ± 6 × 42 ± 9
	TiO_2_ ^a^	Elongated	7 ± 2 × 66 ± 6	9 ± 2 × 66 ± 7	9 ± 3 × 67 ± 9	8 ± 2 × 70 ± 7
	TiO_2_ ^b^	Elongated	10 ± 2 × 68 ± 6	9 ± 3 × 64 ± 7	9 ± 3 × 64 ± 5	9 ± 2 × 67 ± 8
SUN2	TiO_2_ ^a^	Angular	30 ± 4 × 33 ± 7	30 ± 4 × 34 ± 5	27 ± 6 × 35 ± 6	29 ± 4 × 37 ± 4
	TiO_2_ ^b^	Angular	31 ± 7 × 36 ± 7	29 ± 6 × 32 ± 8	31 ± 4 × 35 ± 8	32 ± 4 × 38 ± 8
SUN3	TiO_2_ ^a^	Angular	21 ± 5 × 25 ± 5	22 ± 4 × 25 ± 5	22 ± 4 × 28 ± 5	21 ± 6 × 26 ± 5
	TiO_2_ ^b^	Angular	22 ± 5 × 28 ± 6	27 ± 4 × 25 ± 7	26 ± 5 × 26 ± 6	23 ± 5 × 26 ± 6

**Table 2 nanomaterials-11-02537-t002:** The concentration of Si released from SUN1–3, CA1, and SK1.

Sample Name	Release Amount of Si% (*w*/*w*)	
	Light Condition	Dark Condition
SUN1	10.1 ± 1.4	11.2 ± 2.5
SUN2	5.4 ± 0.9	5.7 ± 0.8
SUN3	3.7 ± 0.4	3.4 ± 0.4
CA1	23 ± 2	19 ± 6

## Data Availability

The data presented in this study are available on a reasonable request from the corresponding author.

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
