# Peer review of "Assessment of Nanopollution from Commercial Products in Water Environments"

_nanomaterials, 2021, doi:10.3390/nano11102537_

Round 1

Reviewer 1 Report

The study is interesting in nature and useful to have a better understanding about the actual effects of the ENMs in the environment. The manuscript can be recommended for publication after inclusion the following modifications:

  1. I expected to see the paper in the format of the journal. Please do that.
  2. There is a lack of information regarding the methodologies used in the study in the
  3. In section 2.1, there is a need to specify all the protocols and standards used for this study.
  4. No characterizations such as XRD have been presented in the manuscript. How the authors confirm the composition of the compounds without such characterizations?
  5. The authors are invited to revise the conclusion by including the numbers in conclusion and abstract of the manuscript to represent better the findings of the research.

Author Response

Comments

Authors response

1.      I expected to see the paper in the format of the journal. Please do that.

The whole paper has been formatted to the journal format

2.      There is a lack of information regarding the methodologies used in the study in the

The comment was unclear and the authors were unable to respond to it, however, the methods used are listed in Sections 2.1 to 2.3.

3.      In section 2.1, there is a need to specify all the protocols and standards used for this study

All modified protocols used to release ENMs from NEPs are listed in Section 2.1.1 (SUN1, SUN2, SUN3, and CA1) to Section 2.1.2 (SK1 and SAN1).

The standards used for method validation are described in lines 179 to 182.

4.      No characterizations such as XRD have been presented in the manuscript. How the authors confirm the composition of the compounds without such characterizations?

All samples investigated in this study were liquids, and thus XRD was not used. The elemental composition was determined from EDX analysis.

5.      The authors are invited to revise the conclusion by including the numbers in conclusion and abstract of the manuscript to represent better the findings of the research.

The abstract and conclusion have been revised.

Reviewer 2 Report

The comments are indexed to the pdf.

Author Response

Comments

Authors response

1.      I wonder if the developed methodology is appropriate or different approach have to be employed. In addition, I think the analysis methods have to be deeply explained and the results could be a little more “squeezed”.

All protocols used in the current study were adopted from previously published methods for specific NEPs i.e. sunscreens and body cream ([43].), socks ([44,45]), and sanitiser ([46][47]. So the current study did not essentially developed new release methods.  The authors were unable to “squeeze” the results section as no specifics were given on what the reviewer deemed can be removed.

2.      Please, try to use fewer abbreviations and acronyms, it is difficult to read the work trying to think of all those acronyms.

The reduction of abbreviations has been effected. MQ, FW, SS, SWP, TW, SDS, PR-ENMs, P-ENMs are respectively replaced with Milli-Q water, Freshwater, seawater, swimming pool water, tap water, sodium dodecyl sulfate, product released ENMs, and pristine ENMs.

3.      Line 36. “Increasing … increases…”. You should be modify the redaction

The sentence was revised.

4.      Line 39. You should write the numbers in the same manner always: in line 29 you write “5 000 NEPs”

The format of the numbers is revised and consistent. Line 31 and Line 42.

5.      line 39 you write “1700 t/annually”. Adapt the format of the numbers.

Number format is revised and consistent. Line 31 and Line 42.

6.      Line 40. It seems that something missing in brackets “(nAg, nTiO2 and ???)”.

Revised. Line 43.

7.      I know the forced release studies are more realistic, but I wonder if it would be better to propose a method for releasing nanoparticles closer to reality shortened. Or is it your intention to liken it to a more real situation? I do not believe that the method used reproduces properly a real situation. Please explain.

The authors agree about the need for heightened realism in such procedures and also standardization. However, for now there are limitations with respect to analytical capability which limits refined developments of procedures, hence for now the authors opted for methods that have been applied elsewhere.

For future studies, we do propose that with advancing/increasing analytical capability, studies should have the indicated complexity. The method limitation and future method recommendation are addressed in Section 2.1 (Line 107 to 116).

8.      Line 111. What kind of foil? In addition, I think starting a sentence with a number is not appropriate, could you rewrite it?

Heavy aluminum duty foil was used (type of foil included in Line 121-2) and the sentence is revised.

9.      Line 112. Please, retype the product symbol “.” as “·”

Changed as per reviewer comment

10.   Line 119. I think that before “suspension” you want to type “.” instead of “;”.

Sentence revised and further description in next sentence given

11.   Line 124. How has it been experimentally confirmed that they contain nanoparticles? Some evidence of these experiments could be included, albeit in supplementary material

The refence ([41]) containing all the evidence of ENMs incorporation is included, Line 136.

12.   I think the authors must add more information about the instrumentation employed, for instance, EDX analyzer characteristics should be included

More information added. Line 157 to 151

13.   Lines 158-160. Was the mixture (H2O2 + HF) added for the samples with Ti or only the HF? It is not entirely clear, you could rewrite the sentence

Sentence revised, Line 175 to 177.

14.   Lines 182. Before “Figures”, release “.”.

Dot removed

15.   Line 185. Could you rewrite this phrase?

Sentence revised, Line 205-206

16.   Lines 194-195. This explanatory phrase could be removed or included above for ease of understanding. Even so, a number of conditions must be met for two EDX spectra to be comparable.

Paragrapah was moved above Figure 1 and revised, Line 202 to 207.

17.   Lines 206-219. How did you proceed to measure the sizes of the particles? I imagine that through IT tools of the TEM itself, but what was the number of particles that were measured to obtain that mean value with its corresponding error

The particle size of all PR-ENMs were measured using Image J software, the mean and sd were calculated from the total measured number.

The procedure to measure particle size is described in Section 2.2.1, Line 161-163.

18.   Line 220. Release “.” before “media”.

Revised, Line 247.

19.   Figure 3. Why purple points are shifted to the right?

Figure fixed.

20.   Lines 258. Type “also” instead of “Also”.

Sentence revised, Line 290.

21.   Line 271. Replace references [67, 68] to the final of the sentence. Instead the references, write “papers”, “works” or something like that.

Sentence revised, Line 305-306.

22.   Line 273. When you say “ultraviolet” you mean “ultraviolet radiation”? If so, it should be reflected in the text.

Sentence revised, Line 307.

23.   Line 288. I don’t understand the last part of the sentence. Please, rewrite.

Sentence revised, Line 325-326.

24.   The last paragraph concludes that these materials can potentially be a source of nanocontamination. However, I have serious doubts, and I think reasonable ones, that these methods of forced liberation reproduce a real situation.

Different methods in other studies have reported occurrence of nanopollution from similar type products. For instance for sunscreens; studies around swimming beaches and swimming pools have demonstrated nanopollution (https://doi.org/10.1016/j.jhazmat.2016.05.099; https://doi.org/10.1016/j.envpol.2011.03.003). Similarly, from laundry washing with washing machineshttps://doi.org/10.1016/j.envint.2011.03.006, nanopollution linked to NEPs use has been reported (https://doi.org/10.1021/es405596y; https://doi.org/10.1016/j.scitotenv.2020.140845; https://doi.org/10.1016/j.scitotenv.2019.136010). So NEPs applied topically exhibit considerable release likelihood, so are those from ENMs impregnated fabrics. So inspite of differing methods; the product sample is relevant for real life nanopollution

25.   Line 331. Why you reflect these two elements (Ti, Zn) and the Zn2+ cation? When you refer the two elements, you mean particulate form (ZnO)? I don´t know how you can distinguish between cationic and oxide form

The authors did not clearly understand the comment. However, the study quantified the total concentration of Ti (from nTiO2), Zn (from nZnO), Zn2+ (from dissolution of nZnO), and Ag+ (from dissolution of nAg). Zn2+ and Ag+ were isolated from release media before sample digestion. The procedure is described in Line 148-154.

26.   Line 458. LOD for Al is 10 ppb? It seems a little high…

Yes, the operational LOD was at 10 ppb as sensitivity stability and reproducibility challenges were experienced at lower than 10.

27.   The conclusions could have more impact on some key aspects, but are generally well written.

The section has been revised.

Round 2

Reviewer 2 Report

The paper has been improved. Congratulations!